# Large Language Models-guided Dynamic Adaptation for Temporal Knowledge Graph Reasoning

Jiapu Wang[1], Kai Sun[1,*] Linhao Luo[2], Wei Wei[3], Yongli Hu[1,†], Alan Wee-Chung Liew[4],
Shirui Pan[4,†] Baocai Yin[1]

[1]Beijing University of Technology, China,    [2]Monash University, Australia
[3]University of Hong Kong, China,    [4]Griffith University, Australia
{jpwang, sunkai}@emails.bjut.edu.cn, linhao.luo@monash.edu, weiwei1206cs@gmail.com
{huyongli, ybc}@bjut.edu.cn, {a.liew, s.pan}@griffith.edu.au

## Abstract

Temporal Knowledge Graph Reasoning (TKGR) is the process of utilizing temporal information to capture complex relations within a Temporal Knowledge Graph (TKG) to infer new knowledge. Conventional methods in TKGR typically depend on deep learning algorithms or temporal logical rules. However, deep learning-based TKGRs often lack interpretability, whereas rule-based TKGRs struggle to effectively learn temporal rules that capture temporal patterns. Recently, Large Language Models (LLMs) have demonstrated extensive knowledge and remarkable proficiency in temporal reasoning. Consequently, the employment of LLMs for Temporal Knowledge Graph Reasoning (TKGR) has sparked increasing interest among researchers. Nonetheless, LLMs are known to function as black boxes, making it challenging to comprehend their reasoning process. Additionally, due to the resource-intensive nature of fine-tuning, promptly updating LLMs to integrate evolving knowledge within TKGs for reasoning is impractical. To address these challenges, in this paper, we propose a **L**arge **L**anguage **M**odels-guided **D**ynamic **A**daptation (LLM-DA) method for reasoning on TKGs. Specifically, LLM-DA harnesses the capabilities of LLMs to analyze historical data and extract temporal logical rules. These rules unveil temporal patterns and facilitate interpretable reasoning. To account for the evolving nature of TKGs, a dynamic adaptation strategy is proposed to update the LLM-generated rules with the latest events. This ensures that the extracted rules always incorporate the most recent knowledge and better generalize to the predictions on future events. Experimental results show that without the need of fine-tuning, LLM-DA significantly improves the accuracy of reasoning over several common datasets, providing a robust framework for TKGR tasks[3].

## 1 Introduction

Temporal Knowledge Graphs (TKGs) [1, 2] are the structured representations of the real world, which incorporate the temporal dimension to analyze how relations between entities evolve over time. Temporal Knowledge Graph Reasoning (TKGR) focuses on leveraging historical information within TKGs to forecast future events. Prior research [3, 4] on TKGR has primarily relied on temporal logical rules [5] or deep learning algorithms, such as graph neural networks [6–8] and reinforcement learning techniques [9]. However, the deep learning-based TKGRs often suffer from the lack of

---

*Equally Important

†Corresponding authors

[3]Code and data are available at: https://github.com/jiapuwang/LLM-DA.git

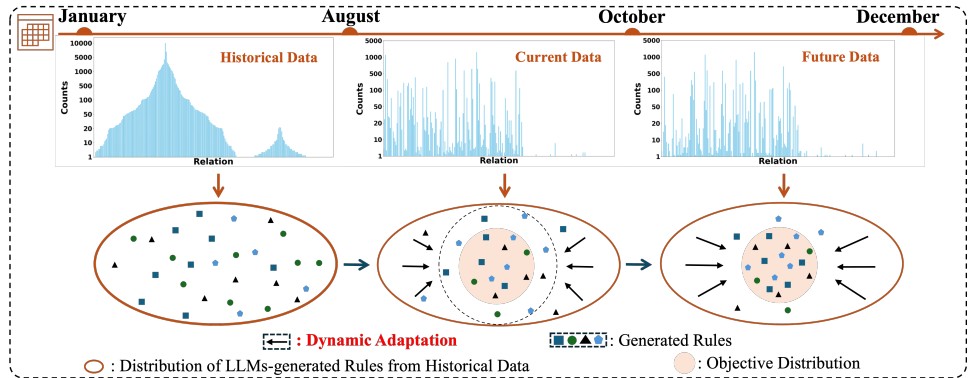

Figure 1: A brief description of LLM-DA. Specifically, LLM-DA harnesses LLMs to formulate general rules on historical data. Subsequently, LLM-DA dynamically guides the LLMs to update these rules based on current data, ensuring they more accurately reflect the objective distribution.

interpretability [10] and are difficult to dynamically update to accommodate new data in TKGs. While rule-based methods offer great interpretability and flexibility, effectively learning temporal logical rules and adapting them to new knowledge remains a huge challenge.

Large Language Models (LLMs) [11], pretrained on large-scale text corpora, have exhibited extensive knowledge and reasoning ability. LLMs is able to effectively grasp intricate semantic and logical relationships within natural language, making them show remarkable performance across a wide range of tasks [12–14]. Recently, LLMs have also demonstrated surprising ability in temporal reasoning [15–17]. By utilizing their powerful contextual processing and pattern recognition abilities, LLMs can extract meaningful temporal patterns and complex temporal dependencies from historical data within TKGs [17–19], thereby significantly enhancing their temporal reasoning capabilities. Thus, leveraging the capabilities of LLMs [20] holds great promise for enhancing the performance of TKGR tasks.

Previous research on LLMs for TKGRs has primarily focused on prompting the LLMs with historical events and asking the LLMs to infer new facts [21–23]. Despite these accomplishments, LLMs are known to be black boxes, leaving it unclear which temporal patterns contribute to the reasoning results. Besides, LLMs suffer from the issue of hallucinations [24], which further undermines the trustfulness of the results. Moreover, it is impractical to promptly update LLMs to incorporate the evolving knowledge within TKGs for reasoning.

Due to the evolving nature, the knowledge within TKGs would continuously update over time, which results in a temporal distribution shift from the initial observations to the future facts [25, 26]. As illustrated in Figure 1, the distribution of the relations in TKG changes dramatically over longer intervals. Despite LLMs possessing abundant knowledge via pre-training, it is still essential to accommodate up-to-date knowledge for reasoning. However, continually updating LLMs is highly impractical due to the intensive resources required for fine-tuning [27]. Additionally, TKGs usually contain significant noise, necessitating an efficient process to extract relevant information for LLMs to discern the underlying temporal patterns.

To address these challenges, this paper proposes a **L**arge **L**anguage **M**odel-guided **D**ynamic **A**daptation (LLM-DA) method for TKGR tasks, which dynamically adapts to the new knowledge and conduct interpretable reasoning powered by LLMs. Specifically, LLM-DA leverages the capabilities of LLMs to analyze historical data and extract temporal logical rules, unveiling temporal patterns and facilitating interpretable reasoning. To efficiently adapt to the new distribution of TKGs, LLM-DA introduces an innovative dynamic adaptation strategy. This strategy iteratively updates the rules generated by LLMs instead of updating the LLMs themselves with the latest events. The extracted rules are dynamically updated and ranked to ensure they consistently incorporate the most recent knowledge and improve predictions for future events, all without the resource-intensive process of LLM fine-tuning.

In order to facilitate the rule generation and update, LLM-DA employs a contextual relation selector to meticulously filter the relations in TKG. The selector identifies the top-$k$ most important relations for each rule head based on their semantic similarities. For example, given a rule head "`president_of`", the relevant relations might be "`occupation_of`" and "`politician_of`". These selected relations are fed into the LLMs as context to ensure LLMs are aligned with the temporal data, enhancing their abilities in uncovering the underlying temporal patterns.

The main contributions of this paper are summarized as follows:

- This paper attempts to harness the ability of Large Language Models (LLMs) for rule-based Temporal Knowledge Graph Reasoning (TKGR) to unveil temporal patterns and facilitate interpretable reasoning;

- This paper proposes an innovative dynamic adaptation strategy that iteratively updates the LLM-generated rules with the latest events, allowing for better adaptation to the constantly changing dynamics within TKGs;

- This paper introduces the contextual relation selector to identify the top $k$ relevant relations, ensuring higher contextual relevance and enhancing the ability of LLMs to understand complex temporal patterns;

- Experimental results on several widely used datasets show that LLM-DA significantly enhances TKGs reasoning accuracy without requiring fine-tuning LLMs.

## 2 Related Work

### 2.1 Temporal Knowledge Graph Reasoning

Temporal Knowledge Graph Reasoning (TKGR) [28–34] aims to leverage historical information within TKGs to forecast future events, which can be roughly categorized into two groups: Rules-based TKGR methods and Deep learning-based TKGR methods.

**Rules-based TKGR methods** [35] enhance TKGs inference by leveraging temporal logical rules to accurately predict future events. TLmod [36] introduces a sophisticated pruning strategy to derive rules and selects the high-confidence rules for TKGR tasks. TLogic [5] learns temporal logical rules from TKGs based on temporal random walks, and subsequently feeds these rules into a symbolic reasoning module for predicting future events. TILP [37] proposes a differentiable framework for temporal logical rule learning, utilizing constrained random walks to enhance the learning process. TFLEX [38] advances beyond learning simplistic chain-like rules by proposing a temporal feature-logic embedding framework. Although temporal logical rules can reveal hidden temporal patterns within TKGR, effectively extracting these rules from TKG still remains a significant challenge.

**Deep learning-based TKGR methods** [39–42] employ the deep learning techniques to capture the hidden temporal patterns to predict the future events in TKGs. RE-NET [43] uses a recurrent event encoder and a neighborhood aggregator to encode historical facts and model their connections, enhancing future event predictions in TKGs. Based on RE-NET, RE-GCN [44] further employs RGCN and GRU to aggregate neighboring messages and model the temporal dependency. CyGNet [45] employs a copy-generation mechanism for capturing global repetition frequencies. TiRGN [46] integrates both local and global historical data to capture the sequential, repetitive, and cyclical patterns inherent in historical data. However, the deep neural networks adopted by these methods often lack interpretability, making it difficult to verify the predictions.

### 2.2 Large Language Models for TKGR

Large Language Models (LLMs) for TKGR generally leverage the sufficient knowledge and reasoning ability of LLMs to conduct reasoning on TKGs. TIMEBENCH [47] proposes a comprehensive hierarchical temporal reasoning benchmark to provide a thorough evaluation for investigating the temporal reasoning capabilities of LLMs. Luo *et al.* [23] performs fine-tuning on known data and then leverage a sequence of established factual information to predict and generate the subsequent event in the series. PPT [21] converts the TKGC task into a masked token prediction task using a Pre-trained Language Model and designs specific prompts for various types of intervals between timestamps to enhance the extraction of semantic information from temporal data. GPT-NeoX [48] implements a

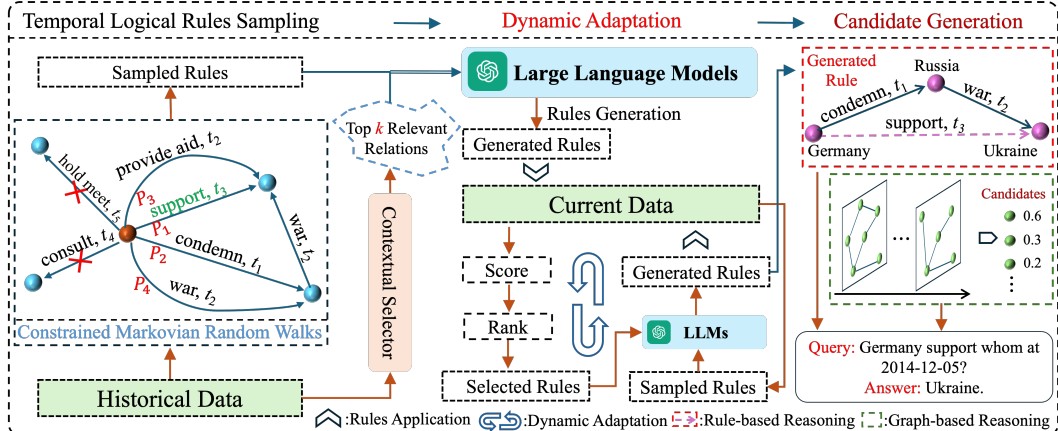

Figure 2: The Framework of LLM-DA. Specifically, LLM-DA first analyzes historical data to extract temporal rules and utilizes the powerful generative capabilities of LLMs to generate general rules. Subsequently, LLM-DA updates these rules using current data. Finally, the updated rules are applied to predict future events.

method that utilizes GPT for TKGs forecasting through in-context learning without any fine-tuning. Similarly, Mixtral-8x7B-CoH [22] also does not require fine-tuning and adopts a "chain-of-history" reasoning method to effectively generate high-order historical information step-by-step. Nevertheless, existing methods only prompt the LLMs with historical events from TKG, which are limited by the quality of input data and interpretability of LLMs.

Different from the aforementioned LLM for TKGRs, the proposed LLM-DA is LLMs for rule-based TKGR method. By utilizing explicit rules, LLM-DA ensures the interpretability of the LLM-generated processes and dynamically updates the rules to adapt to new data, thereby addressing a major limitation of LLMs-enhanced deep learning-based TKGRs.

## 3 Preliminary

**Temporal Knowledge Graph (TKG).** TKG $\mathcal{G} = \{\mathcal{E}, \mathcal{R}, \mathcal{T}, \mathcal{Q}\}$ is a collection of entity set $\mathcal{E}$, relation set $\mathcal{R}$ and timestamp set $\mathcal{T}$. Specifically, each quadruplet is denoted as $(e_s, r, e_o, t) \in \mathcal{Q}$, where $e_s, e_o \in \mathcal{E}$ represent the entities, $r \in \mathcal{R}$ denotes the relation and $t \in \mathcal{T}$ is the timestamp.

**Temporal Logical Rule.** Temporal logical rules $\rho$ define the relation between two entities $e_s$ and $e_o$ at timestamp $t_l$,

$$\rho := r(e_s, e_o, t_l) \leftarrow \wedge_{i=1}^{l-1} r^*(e_s, e_o, t_i), \tag{1}$$

where the left-hand side denotes the rule head with relation $r$ that can be induced by ($\leftarrow$) the right-hand rule body. The rule body is represented by the conjunction ($\wedge$) of a series of body relations $r^* \in \{r_1, ..., r_{l-1}\}$ [49].

**Different Data Types.** *Historical data* refers to data that have occurred in the past, reflecting the state of things in the past period of time. *Current data* reflects the latest state of things in the present, from a more recent point in time. *Future data* refers to data that will occur, reflecting the possible future trend of things. Specifically, the historical data, current data and future data correspond to the training, validation, and test datasets of prior research [46, 22].

## 4 Methodology

In this section, we propose a novel **L**arge **L**anguage **M**odel-guided **D**ynamic **A**daptation (LLM-DA) method for TKGR tasks. LLM-DA contains four main stages: **Temporal Logical Rules Sampling** explores the constrained Markovian random walks to extract temporal logical rules from the historical data; **Rule Generation** utilizes the powerful generative capabilities of LLMs to extract

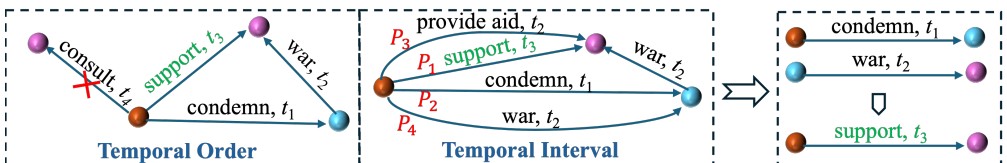

Figure 3: The constraints of the constrained Markovian random walks. "×" denotes this path does not exist, "$P_i$" indicates the transition probability.

meaningful temporal patterns and complex temporal dependencies from historical data within TKGs; **Dynamic Adaptation** leverages LLMs to update the LLM-generated general rules using current data; **Candidate Reasoning** combines rules-based reasoning and graphs-based reasoning to generate candidates. The whole framework is illustrated in Figure 2.

## 4.1 Temporal Logical Rules Sampling

Temporal logical rules sampling is a constrained Markovian random exploration process, which explores the *Timestamp-Entity* joint level weighted temporal random walks. Generally, the constrained Markovian random exploration process is not only strictly constrained by the graph structure but also deeply influenced by the temporal dimension when choosing the next node.

**Constrained Markovian Random Walks.** Compared to traditional Markov random walks, constrained Markovian random walks primarily reflect in two key factors such as temporal order and temporal intervals when choosing the next node. As shown in Figure 3, edges of the next state are selected based on the temporal order, and edges are weighted by the temporal interval. Specifically, edges with shorter interval receive higher weights, thus making the random walk more inclined to choose nodes that are temporally closer.

To maximize the performance of random walks, LLM-DA needs to ensure that the random walks simultaneously satisfy the Markov property and additional constraints. Given the edge $r(e_x, e_y, t_l)$, LLM-DA employs the Markovian random walks to search the closed temporal paths, further obtaining the set of candidates $\mathcal{M}_r$. To ensure the efficiency of our framework, LLM-DA introduces the filtering operator $\chi(t)$, which explores the temporal order to filter these candidate paths. The filtering operator $\chi(t)$ can be denoted as:

$$\chi(t) = \begin{cases} 1, & \text{if } t < t_l, \\ 0, & \text{otherwise.} \end{cases} \tag{2}$$

After filtering these candidate paths, LLM-DA further introduces the temporal interval as the another constraint. This constraint selects the next node based on the transition probability $P$. For a temporal logical rule, the edge $r(e_x, e_y, t)$ represents a connection from node $e_x$ to node $e_y$ at timestamp $t$. Thus, the transition probability of the constrained Markovian random walks is expressed as:

$$P_{xy}(t) = \frac{\exp(-\lambda(t - T))}{\sum_{(e_x, e_y, t') \in \mathcal{G}} \exp(-\lambda(t' - T))}, \tag{3}$$

where $P_{xy}(t)$ is the probability of transitioning from node $e_x$ to node $e_y$ at time $t$, and $w(t) = \exp(-\lambda(t - T))$ denotes an exponential decay function that weights the edges based on the time difference, $T$ is the current time, $t$ is the timestamp of the edge, and $\lambda$ is the decay rate parameter, which controls the weight given to more recent times. In this way, more recent times (i.e., $t$ closer to $T$) are assigned greater weight. The detailed theoretical analysis is shown in Appendix B. Through the constrained Markovian random walks on the historical data, LLM-DA extracts the temporal rules $\mathcal{S}$ from the sampled temporal paths.

## 4.2 Rule Generation

Rule generation typically utilizes the powerful generative capabilities of LLMs to improve the insufficient coverage and low quality of the extracted temporal rules $\mathcal{S}$. Specifically, LLM-DA first employs the contextual relation selector to identify the top $k$ relevant relations. Additonally, LLMs

generally have powerful generative capabilities, allowing them to extract meaningful temporal patterns and complex temporal dependencies from historical data within TKGs. Thus, LLM-DA inputs the extracted rules $\mathcal{S}$ and the top $k$ relevant contextual relations into LLMs to generate high-coverage and high-quality general rules.

Firstly, LLM-DA filters the input contextual information through a contextual relation selector. TKGs generally contain rich temporal relations, requiring the processing of a substantial amount of context when LLMs conduct TKGR tasks. Contextual relation selector aims to filter the relation pool to identify the top $k$ most relevant relations. For a temporal rule $\rho := r(e_s, e_o, t_l) \leftarrow \bigwedge_{i=1}^{l-1} r^*(e_s, e_o, t_i)$, LLM-DA leverages the pre-trained Sentence-Bert [50] to embed rule head $r$ and each of its corresponding candidate relations $c_j$ into a common space, obtaining embedding vectors $\mathbf{c}_j$ and $\mathbf{r}$:

$$\mathbf{c}_j, \mathbf{r} = \text{Sentence-Bert}(c_j, r). \tag{4}$$

Following, LLM-DA calculates the relevance score $S(r, c_j)$ through the cosine similarity function:

$$S(r, c_j) = \frac{\mathbf{r} \cdot \mathbf{c}_j}{\|\mathbf{r}\|\|\mathbf{c}_j\|}, \tag{5}$$

where $\cdot$ denotes the dot product operation, and $\| \cdot \|$ denotes the norm of the vector.

Finally, LLM-DA sorts the relevance scores in descending order and selects the top $k$ candidate relations as effective candidate relations, ensuring that the selected relations are semantically closely aligned with the rule head $r$:

$$\{c_{j_1}, c_{j_2}, \ldots, c_{j_k}\} = \text{Top-}k\{S(r, c_j)\}, \tag{6}$$

where $\{c_{j_1}, c_{j_2}, \ldots, c_{j_k}\}$ are the top $k$ most relevant relations.

After the above contextual relation selector, LLM-DA inputs the extracted temporal rules and the top $k$ most relevant candidate relations $\{c_{j_1}, c_{j_2}, \ldots, c_{j_k}\}$ corresponding to the rule head $r$ into LLMs to achieve the generation of the high-coverage and high-quality general rules $\mathcal{S}_g$. A simple example of prompt is shown in the following prompt box:

---

**Prompt for Rule Generation**

You are an expert in TKGR, and please generate as many temporal logical rules as possible related to '$r$' based on extracted temporal rules.

For the relations in rule body, you are going to choose from the candidate relations: "$\{c_{j_1}, c_{j_2}, \ldots, c_{j_k}\}$".

---

The detailed instruction for *Rule Generation* can be found in the Appendix A.1.

## 4.3 Dynamic Adaptation

Dynamic adaptation generally leverages LLMs to update the LLMs-generated general rules using current data. Due to the evolving nature of TKGs, the LLMs-generated rules $\mathcal{S}_g$ become less suitable for new data, causing high-quality rules to gradually degrade into low-quality rules. To address this, LLM-DA extracts temporal rules from the current data and uses these rules as a standard to update the low-quality rules. This ensures that these rules always incorporate the most recent knowledge and better generalize to the predictions on future events.

As shown in the *Prompt for Dynamic Adaptation* prompt box, the input of LLMs for dynamic adaptation mainly contains two parts, including the update of low-quality rules and extracted rules from current data.

---

**Prompt for Dynamic Adaptation**

You are an expert in TKGR, and please analyze these LLMs-generated rules and update the **low-quality rules** based on the **extracted rules** from current data.

For the relations in rule body, you are going to choose from the candidate relations: "$\{c_{j_1}, c_{j_2}, \ldots, c_{j_k}\}$".

---

*Low-Quality Rules.* As TKGs evolve over time, the LLM-generated rule set $\mathcal{S}_g$ may become increasingly difficult to fit to the current data, eventually turning into low-quality rules. Thus, LLM-DA applies the *Confidence* [51] metric to score each temporal rule $\rho$ on the current data. *Confidence* generally measures the reliability of the temporal rule $\rho$, which can be defined as the proportion of temporal fact pairs that satisfy the rule body $rule\_body(\rho)$ and also satisfy the whole $rule(\rho)$:

$$c_\rho = \frac{\text{Number of temporal fact pairs satisfying } rule\_body(\rho)}{\text{Number of temporal fact pairs satisfying } rule(\rho)}, \qquad (7)$$

where $c_\rho$ denotes the confidence of rule $\rho$. The higher the confidence, the greater the reliability of the rule. In other words, when the rule body is true, the likelihood of the rule head being true is higher. Subsequently, we select the subset of rules with low confidence $\mathcal{S}_{g(\text{low})} = \{\rho \in \mathcal{S}_g \mid c_\rho < \theta\}$, where $\theta$ denotes the threshold of the low confidence.

*Extracted Rules from Current Data.* To address the issue of the broad range of rules generated by LLMs, LLM-DA explores constrained Markovian random walks to extract temporal logical rules from current data, which serves as a standard to constrain the scope of dynamic adaptation. Through iteratively invoking LLMs, the accuracy of LLMs in predicting future events can be enhanced. The detailed prompt for *Dynamic Adaptation* refers to Appendix A.2.

Finally, LLM-DA updates these LLMs-generated low-quality rules through the extracted rules from current data, further obtaining the rules set $\mathcal{S}_d$.

## 4.4 Candidate Reasoning

Candidate reasoning aims to infer potential answers for the query by integrating the above LLMs-generated rules and GNNs-based predictions. Specifically, LLM-DA mainly consists of two key modules: **Rule-based Reasoning** and **Graph-based Reasoning**.

**Rule-based Reasoning.** Rule-based reasoning typically utilizes the above LLMs-generated high-scoring rules to conduct in-depth logical reasoning within TKGs, deducing new entities as potential answers [5]. Given a query $(e_s, r, ?, t_l)$, LLM-DA scores the rule through the Equation 7, and then select the high-scoring rules:

$$\mathcal{S}'_d = \{\rho \mid c_\rho > \gamma, \rho \in \mathcal{S}_d\}, \qquad (8)$$

where $\mathcal{S}_d$ is the rule set obtained after the *Dynamic Adaptation* process, $\mathcal{S}'_d$ is the high-confidence rule set, in which the score $c_\rho$ of the rule $\rho$ is greater than the threshold $\gamma$. Following the rule $\rho \in \mathcal{S}'_d$, we can find the reasoning paths and further derive the entity $e'_o$:

$$(e_s, r, e'_o, t_l) \leftarrow \wedge_{i=1}^{l-1}(e_s, r_i, e'_o, t_i), \qquad (9)$$

where $e'_o$ is the candidate derived based on the rule $\rho$, and $t_{l-1} \geq \cdots \geq t_1$. Considering the time decay property of temporal data, we further select candidates that are most relevant to the query:

$$\text{Score}_{(\rho, e'_o)} = \sum_{\rho \in R'_s} \sum_{body(r)(e_s, e'_o, t_l) \in \mathcal{G}} (c_\rho + \exp(-\lambda(t_l - t_o))), \qquad (10)$$

where $\text{Score}_{(\rho, e'_o)}$ indicates the score of the candidate $e'_o$ obtained through the searched path in TKGs based on rule $\rho$ at the time point $t_o$, $c_\rho$ denotes the confidence of the temporal rule $\rho$, $\lambda$ represents the decay rate, and $body(\rho)(e_s, e'_o, t_l) \in \mathcal{G}$ denotes the path in TKGs that satisfies the rule body.

**Graph-based Reasoning.** Due to inconsistent data distribution, candidates generated solely based on rules may not fully match all query answers. Thus, we introduce the graph-based reasoning function $f_g(Query)$ to further predict the candidates of the query, and the score can be computed through the inner product operation:

$$\text{Score}_{(graph, e'_o)} = \langle f_g(Query), e'_o \rangle. \qquad (11)$$

Since the candidates of *Rule-based Reasoning* $\mathcal{E}_{(\rho, e'_o)}$ and *Graph-based Reasoning* $\mathcal{E}_{(graph, e'_o)}$ have the overlap and difference, we assign the score $\text{Score}_{(\rho, e'_o)}$ as 0 where $e'_o \in \mathcal{E}_{(graph, e'_o)}$, $e'_o \notin \mathcal{E}_{(\rho, e'_o)}$, and vice versa. The whole score of the candidate can be computed as follows:

$$\text{Score}_f = \alpha \cdot \text{Score}_{(\rho, e'_o)} + (1 - \alpha) \cdot \text{Score}_{(graph, e'_o)}, \qquad (12)$$

where $\text{Score}_f$ represents the final score of the candidate $e'_o$ from rule-based reasoning and graph-based reasoning modules, and $\alpha$ is used to assign weights to different scores. Finally, LLM-DA aggregates all candidates, and then sorts these candidates to select those that best meet the query requirements.

Table 1: Link prediction results on ICEWS14 and ICEWS05-15. The best results are in bold and - means the result is unavailable. ♠ denotes the TKGR methods, ♣ represents the LLMs-based TKGR, and LLM-DA (·) indicates replacing the graph-based reasoning module $f_g(Query)$ with TKGRs (♠).

| Type | Models | Train | ICEWS14 | | | | ICEWS05-15 | | | |
|---|---|---|---|---|---|---|---|---|---|---|
| | | | MRR | Hit@1 | Hit@3 | Hit@10 | MRR | Hit@1 | Hit@3 | Hit@10 |
| ♠ | RE-NET | ✓ | 0.388 | 0.290 | 0.436 | 0.576 | 0.441 | 0.332 | 0.512 | 0.650 |
| | RE-GCN | ✓ | 0.425 | 0.320 | 0.476 | 0.627 | 0.478 | 0.371 | 0.535 | 0.682 |
| | TiRGN | ✓ | 0.441 | 0.341 | 0.497 | 0.650 | 0.495 | 0.389 | 0.559 | 0.703 |
| | TLogic | ✓ | 0.390 | 0.295 | 0.437 | 0.573 | 0.459 | 0.360 | 0.518 | 0.646 |
| ♣ | PPT | ✓ | 0.384 | 0.289 | 0.425 | 0.570 | 0.389 | 0.286 | 0.434 | 0.586 |
| | Llama-2-7b-CoH | ✓ | – | 0.349 | 0.470 | 0.591 | – | 0.386 | 0.541 | 0.699 |
| | Vicuna-7b-CoH | ✓ | – | 0.328 | 0.457 | 0.656 | – | 0.392 | 0.546 | 0.707 |
| | GPT-NeoX | ✗ | – | 0.334 | 0.460 | 0.565 | – | – | – | – |
| | Mixtral-8x7B-CoH | ✗ | 0.439 | 0.331 | 0.496 | 0.649 | 0.497 | 0.380 | 0.564 | 0.713 |
| | LLM-DA (RE-GCN) | ✗ | 0.461 | 0.356 | 0.515 | 0.662 | 0.501 | 0.394 | 0.568 | 0.710 |
| | LLM-DA (TiRGN) | ✗ | **0.471** | **0.369** | **0.526** | **0.671** | **0.521** | **0.416** | **0.586** | **0.728** |

## 5 Experiments

### 5.1 Experimental Settings

**Datasets.** ICEWS14 [52] and ICEWS05-15 [52] are the subset of *Integrated Crisis Early Warning System (ICEWS)*, which is a TKG of international political events and social dynamics. ICEWS14 contains events that occurred in 2014, while ICEWS05-15 contains events that occurred between 2005 and 2015. Details of datasets can be referred to Appendix C.1.

**Baselines.** The proposed LLM-DA is compared with several classic TKGR methods, including 1) TKGR methods: RE-NET [43], RE-GCN [44], TiRGN [46] and TLogic [5]; 2) LLMs-based TKGRs: GPT-NeoX [48], Llama-2-7b-CoH, Vicuna-7b-CoH [23], Mixtral-8x7B-CoH [22] and PPT [21]. Here, LLM-DA selects RE-GCN and TiRGN as the graph-based reasoning function $f_g(Query)$. The detail of each baseline is described in Appendix C.2.

**Parameter Setting.** The proposed LLM-DA uses the ChatGPT[4] as the LLM for *Rules Generation* and *Dynamic Adaptation*. LLM-DA chooses *Mean Reciprocal Rank* (MRR) and *Hit@N* ($N = 1, 3, 10$) as evaluation metrics, and presents the *filtered* results (Appendix C.3). Additionally, LLM-DA sets the decay rate $\lambda$ in *Temporal Logical Rules Sampling* and *Candidate Generation*, the threshold $\theta$ in *Dynamic Adaptation*, the min-confidence $\gamma$ and the parameter $\alpha$ in *Candidate Generation* on both datasets as follows: $\lambda = 0.1$, $\theta = 0.01$, $\alpha = 0.9$ and $\gamma = 0.01$, except for $\alpha = 0.8$ on ICEWS05-15. The number of iterations for the *Dynamic Adaptation* is set as 5. All experiments are implemented on a NVIDIA RTX 3090 GPU with i9-10900X CPU.

### 5.2 Performance Comparison

The link prediction experimental results are displayed of ICEWS14 and ICEWS05-15 in Table 1, and the ICEWS18 is shown in Appendix C.4. The experimental analyses are listed as follows:

(1) Experimental results indicate that even without fine-tuning, the proposed LLM-DA can still surpass all LLM-based TKGR methods. This phenomenon demonstrates that the dynamic adaptation strategy can effectively update LLMs-generated general rules with latest events to capture the evolving nature of TKGs, thereby significantly improving the accuracy of future event predictions. Additionally, we present the visualization experiment in Appendix C.6, which validates the superiority of the dynamic adaptation strategy.

(2) Some LLMs-based TKGC methods such as PPT [21] and GPT-NeoX [48], are not always superior to traditional TKGR methods. This is primarily because the rules generated by LLMs are too broad and sometimes fail to precisely adapt to specific data. However, the proposed LLM-DA outperforms the existing state-of-the-art benchmarks on all metrics. This phenomenon proves that LLM-DA can effectively guide LLMs in adjusting rules to the target distribution.

---

[4]We use the snapshot of ChatGPT taken from February 15th 2024 (gpt-3.5-turbo-0215) to ensure the reproducibility.

(3) GPT-NeoX [48] is an important baseline as it also incorporates GPT as an LLM in TKGC tasks. However, LLM-DA shows significant improvement. This phenomenon indicates that the dynamic adaptation strategy can effectively update LLMs-generated rules to adapt to future data.

(4) Furthermore, replacing the graph-based reasoning module $f_g(Query)$ with RE-GCN ("LLM-DA (RE-GCN)") and TiRGN ("LLM-DA (TiRGN)"), the MRR performance shows a slight variation. This variation highlights the importance of incorporating graph-based reasoning function in enhancing the ability to predict future events.

## 5.3 Analysis of Dynamic Adaptation

In LLM-DA, dynamic adaptation aims to continuously update the generated rules to capture temporal patterns and facilitate future predictions. To further investigate its impact, we aim to answer the following questions: **RQ1:** Can the dynamic adaptation better extract the temporal patterns from TKGR for reasoning? **RQ2:** Can the dynamic adaptation adapt to different distributions over time? **RQ3:** Can the iterative dynamic adaptation improve the performance?

Table 2: Ablation study on different data without dynamic adaptation on both datasets. The best results are in bold. "LLM-DA *w* H" indicates "only using the historical data", "LLM-DA *w* C" is "only using the current data", and "LLM-DA *w* H+C" denotes "using the historical and current data".

| Models | ICEWS14 | | | | ICEWS05-15 | | | |
|---|---|---|---|---|---|---|---|---|
| | MRR | Hit@1 | Hit@3 | Hit@10 | MRR | Hit@1 | Hit@3 | Hit@10 |
| LLM-DA (TiRGN) *w* H | 0.450 | 0.345 | 0.502 | 0.656 | 0.503 | 0.395 | 0.563 | 0.709 |
| LLM-DA (TiRGN) *w* C | 0.454 | 0.350 | 0.507 | 0.657 | 0.508 | 0.400 | 0.570 | 0.714 |
| LLM-DA (TiRGN) *w* H+C | 0.457 | 0.352 | 0.510 | 0.659 | 0.511 | 0.402 | 0.573 | 0.718 |
| LLM-DA (TiRGN) | **0.471** | **0.369** | **0.526** | **0.671** | **0.521** | **0.416** | **0.586** | **0.728** |

**RQ1:** The ablation study on different data without dynamic adaptation aims to evaluate the impact of dynamic adaptation on performance. Dynamic adaptation typically employs the current data to update the rules generated by LLMs based on historical data. To demonstrate the superiority, we compare three variations, including "*LLM-DA w H*", "*LLM-DA w C*" and "*LLM-DA w H+C*", and the experimental results are shown in Table 2. Specifically, the "*LLM-DA w H+C*" outperforms "*LLM-DA w H*" and "*LLM-DA w C*", indicating that large-scale historical data can provide general knowledge, while current data offers relevant knowledge. The combination of both enhances the prediction of future events. Furthermore, compared to "*LLM-DA w H+C*", LLM-DA shows a significant improvement. This phenomenon demonstrates that the dynamic adaptation strategy can effectively integrate historical data and current data, leveraging the current data to update the rules generated by LLMs on historical data.

**RQ2:** The temporal data has the time decay property, causing the issue of distributional shift. Specifically, the longer the time interval between the data, the more pronounced the shift becomes. To verify whether the dynamic adaptation can adapt to different distributions over time, we conduct the time interval segmented prediction experiment, which typically segments the future data into multiple time intervals based on chronological order, and then conducts the link prediction experiment for each time interval. As shown in Figure 4, the proposed LLM-DA exhibits a significant performance improvement in the MRR metric over RE-GCN and TiRGN in each time interval. This indicates that the proposed LLM-DA can accurately capture the temporal dependencies in TKGs and adapt to the continuously changing temporal data. Furthermore, in Appendix C.5, we conduct long-term horizontal link prediction to forecast events occurring at future time points.

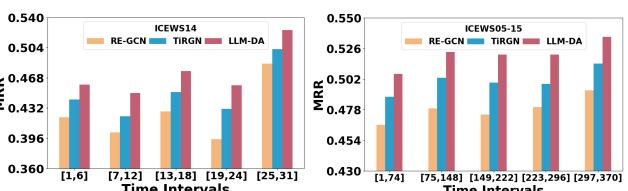

Figure 4: Time interval segmented prediction: MRR performance on both datasets compared to different baselines.

**RQ3:** To further verify whether the number of iterations of the dynamic adaptation module affects the performance, we conduct the different numbers of iterations experiment on both datasets. As shown in Figure 5, the MRR performance exhibits an increasing trend as the number of iterations

increases. These observations indicate that the dynamic adaptation strategy can continuously update the LLMs-generated rules with the latest events through iterations, thereby better adapting to the dynamic changes of TKGs. This further demonstrates the effectiveness and necessity of the dynamic adaptation strategy in handling the evolving nature of TKGs. Moreover, we conduct the parameter analysis experiment to validate the impact of the weight $\alpha$ in Appendix C.7.

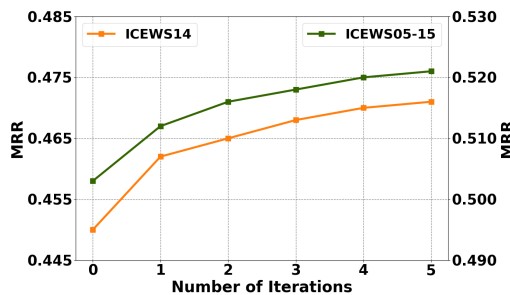

Figure 5: Comparison with different number of iterations on both datasets.

## 6 Conclusion

In this paper, we propose a novel **L**arge **L**anguage **M**odel-guided **D**ynamic **A**daptation (LLM-DA) method to enhance TKGR tasks. Specifically, LLM-DA leverages a contextual relation selector to identify the top $k$ most relevant relations, thereby selecting pertinent contextual information. Subsequently, LLM-DA harnesses the generative capabilities of LLMs to analyze historical data and derive general rules. Furthermore, LLM-DA proposes a dynamic adaptation strategy to update the LLM-generated rules with latest events, further capturing the evolving nature of TKGs. Experimental results on several datasets unequivocally demonstrate that LLM-DA achieves competitive performance compared to state-of-the-art methods. Appendix D further analyzes the limitations of LLM-DA and provides an outlook for future work.

## Acknowledgment

This work was funded by the National Key R&D Program of China (No. 2021ZD0111902), National Natural Science Foundation of China (No. U21B2038), R&D Program of Beijing Municipal Education Commission (KZ202210005008). The first author is funded by the China Scholarship Council (CSC) from the Ministry of Education, China.

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

# Appendix

## A    Prompt for LLMs

### A.1    Prompt for Rule Generation

---

**Prompt for Rule Generation**

Temporal Logical Rules "$R_l(X, Y, T_l) \leftarrow \wedge_{i=1}^{l-1} R_i(X,\ Y,\ T_i)$" typically describe how the relation '$R_l$' between entities '$X$' and '$Y$' evolves from past time steps '$T_i$ $(i = \{1, \cdots, (l-1)\})$' (Rule body) to the next timestamp '$T_l$' (Rule head), and please follow the constraint "$T_1 \leq \cdots \leq T_{l-1} < T_l$".

You are an expert in temporal knowledge graph reasoning, and please generate as many temporal logical rules as possible related to '$R_l$' based on extracted temporal rules.

Here are a few examples:
**Example 1:**
    Rule Head:
        Cooperate_economically $(X, Y, T)$
    Extracted Rules:
        Cooperate_economically $(X, Y, T_2) \leftarrow$ Provide_aid $(X, Y, T_1)$
        Cooperate_economically $(X, Y, T_3) \leftarrow$ Host_a_visit $(X, Z_1, T_1)$ & Negotiate $(Z_1, Y, T_2)$
        ...
    Generated Temporal Logical Rules:
        Cooperate_economically $(X, Y, T_2) \leftarrow$ Engage_in_negotiation $(X, Y, T_1)$
        Cooperate_economically $(X, Y, T_3) \leftarrow$ inv_Engage_in_negotiation $(X, Z_1, T_1)$ & Make_a_visit $(Z_1, Y, T_2)$
        ...

**Example 2:**
    Rule Head:
        Appeal_for_economic_aid $(X, Y, T)$
    Extracted Rules:
        Appeal_for_economic_aid $(X, Y, T_2) \leftarrow$ inv_Reduce_or_stop_military_assistance $(X, Y, T_1)$
        Appeal_for_economic_aid $(X, Y, T_3) \leftarrow$ inv_Express_intent_to_cooperate $(X, Z_1, T_1)$ & Make_statement $(Z_1, Y, T_2)$
        ...
    Generated Temporal Logical Rules:
        Appeal_for_economic_aid $(X, Y, T_2) \leftarrow$ Make_an_appeal_or_request $(X, Y, T_1)$
        Appeal_for_economic_aid $(X, Y, T_3) \leftarrow$ inv_Make_an_appeal_or_request $(X, Z_1, T_1)$ & Make_statement $(Z_1, Y, T_2)$
        ...

**Extracted Rules from Historical Data:**
    ......

Let's think step-by-step, please generate as many as possible most relevant temporal rules that are relative to "{head_rule$(X, Y, T)$}" based on the above extracted rules from historical data.
    For the relations in rule body, you are going to choose from the candidate relations: "$\{c_{j_1}, c_{j_2}, \ldots, c_{j_k}\}$".

Return the rules only without any explanations.

---

## A.2 Prompt for Dynamic Adaptation

---

**Prompt for Dynamic Adaptation**

Temporal Logical Rules "$R_l(X, Y, T_l) \leftarrow \wedge_{i=1}^{l-1} R_i(X, Y, T_i)$" typically describe how the relation '$R_l$' between entities '$X$' and '$Y$' evolves from past time steps '$T_i$ ($i = \{1, \cdots, (l-1)\}$)' (Rule body) to the next timestamp '$T_l$' (Rule head), and please follow the constraint "$T_1 \leq \cdots \leq T_{l-1} < T_l$".

You are an expert in temporal knowledge graph reasoning, and please analyze these LLMs-generated rules and update the low-quality rules based on the extracted rules from current data.

Here are a few examples:
**Example 1:**
    Rule Head:
        inv_Provide_humanitarian_aid ($X, Y, T$)
    Low Quality Temporal Logical Rules:
        Make_a_visit ($X, Y, T_2$) ← Provide_military_protection_or_peacekeeping ($X, Y, T_1$)
        Make_a_visit ($X, Y, T_4$) ← Appeal_for_diplomatic_cooperation_(such_as_policy_support) ($X, Z_1, T_1$) & inv_Consult ($Z_1, Z_2, T_2$) & inv_Make_statement ($Z_2, Y, T_3$)
    Generated High Quality Temporal Logical Rules:
        Make_a_visit ($X, Y, T_2$) ← Express_intent_to_meet_or_negotiate ($X, Z_1, T_1$) & Make_a_visit ($Z_1, Y, T_2$)
        Make_a_visit ($X, Y, T_3$) ← Consult ($X, Z_1, T_1$) & Engage_in_negotiation ($Z_1, Z_2, T_2$) & Make_a_visit ($Z_2, Y, T_3$)
        ...
**Example 2:**
    Rule Head:
        inv_Provide_humanitarian_aid ($X, Y, T$)
    Low Quality Temporal Logical Rules:
        inv_Provide_humanitarian_aid ($X, Y, T_2$) ← inv_Investigate ($X, Y, T_1$)
        inv_Provide_humanitarian_aid ($X, Y, T_2$) ← inv_Engage_in_diplomatic_cooperation ($X, Y, T_1$)
    Generated High Quality Temporal Logical Rules:
        inv_Provide_humanitarian_aid ($X, Y, T_2$) ← inv_Provide_aid ($X, Y, T_1$)
        inv_Provide_humanitarian_aid ($X, Y, T_3$) ← Criticize_or_denounce ($X, Z_1, T_1$) & Sign_formal_agreement ($Z_1, Y, T_2$)
        ...

**Low-quality Temporal Logical Rules:**
    ......

**Extracted Rules from Current Data:**
    ......

Let's think step-by-step, and please update the low-quality temporal logic rules related to "$\{head\_rule(X, Y, T)\}$" based on the extracted rules from current data.
    For the relations in rule body, you are going to choose from the candidate relations: "$\{c_{j_1}, c_{j_2}, \ldots, c_{j_k}\}$".

Return the rules only without any explanations.

---

# B Theoretical Analysis of the Constrained Markovian Random Walks

In conducting a theoretical analysis of constrained Markovian random walks, we can examine their characteristics and effects from several key aspects:

- **Impact of Time Weighting:** By using weighted transition probabilities $P_{xy}(t_l)$, constrained Markovian random walks can assign different importance to neighboring nodes at different time points. In particular, through the exponential decay function $w(t) = \exp(-\lambda(t_l - T))$, we can control the extent to which time influences the transition probabilities. The larger the parameter $\lambda$, the stronger the preference for recent events, which helps capture short-term dynamic changes.

- **Traversal Properties:** An important feature of constrained Markovian random walks is their ability to traverse different paths in TKGs. Since the transition probabilities consider time information, constrained Markovian random walks are more likely to explore paths with temporal continuity compared to traditional random walks, thereby better reflecting temporal relationships.

# C Experiments

## C.1 Datasets

Table 3 provides a comprehensive overview of the statistical information for the entire datasets used in our experiments. This includes key metrics such as the number of entities, the number of relations, the number of temporal facts, and the granularity of each dataset. This statistical information is crucial for understanding the scale and complexity of the datasets, as well as for evaluating the performance of LLM-DA under different experimental conditions.

*Integrated Crisis Early Warning System (ICEWS)* is a TKG for international crisis early warning and analysis, which collects and integrates data on political events, social dynamics, and international relations worldwide. Specifically, ICEWS14 [52] contains events that occurred from $1/1/2014$ to $12/31/2014$; ICEWS05-15 [52] includes events that occurred between the years 2005 and 2015; ICEWS18 [22] covers events from $1/1/2018$ to $10/31/2018$.

Table 3: Statistic information of whole datasets.

| Datasets | #Entities | #Relations | #Historical Data | #Current Data | #Future Data | #Granularity |
|---|---|---|---|---|---|---|
| ICEWS14 | 6,869 | 230 | 74,845 | 8,514 | 7,371 | 24 hours |
| ICEWS05-15 | 10,094 | 251 | 368,868 | 46,302 | 46,159 | 24 hours |
| ICEWS18 | 23,033 | 256 | 373,018 | 45,995 | 49,545 | 24 hours |

## C.2 Baselines

The proposed model LLM-DA is compared with some classic TKGR methods, including TKGR methods and LLMs-based TKGR methods.

**TKGR methods**

- RE-NET [43]: RE-NET first TKGC method for predicting future events, which employs a recurrent event encoder and a neighborhood aggregator to encode historical facts;
- RE-GCN [44]: RE-GCN is an extension of RE-NET, which introduces RGCN and GRU to encode historical facts;
- TiRGN [46]: TiRGN employs constrained random walks for temporal logical rule learning;
- TLogic [5]: TLogic learns the temporal logical rules from TKGs based on temporal random walks;

**LLMs-based TKGR methods**

- GPT-NeoX [48]: GPT-NeoX introduces in-context learning with GPT for TKG forecasting;

- Llama-2-7b-CoH, Vicuna-7b-CoH [23]: Llama-2-7b-CoH, Vicuna-7b-CoH fine-tune Llama-2-7b and Vicuna-7b on historical data for TKG prediction;
- Mixtral-8x7B-CoH [22]: Mixtral-8x7B-CoH leverage Mixtral-8x7B to explore high-order histories step-by-step for the TKG prediction;
- PPT [21]: PPT introduces the Pre-trained Language Model to predict future events.

## C.3 Link Prediction Metrics

We evaluate our model using Mean Reciprocal Rank (MRR) and Hit@$N$ metrics, where $N$ is set to 1, 3, and 10. Higher scores indicate better performance. Lastly, we present the final experimental results, termed as *filtered*, which exclude all corrupted quadruplets from the TKG.

- Mean Reciprocal Rank (MRR): MRR measures the average accuracy of ranking predictions made by a model. It is calculated by taking the reciprocal of the rank of the correct answer for each query and then averaging these reciprocal ranks across all queries. Mathematically, for each query, if the rank of the correct answer is $r$, then the MRR score for that query is $\frac{1}{r}$. The overall MRR score is the average of these reciprocal ranks. MRR values range between 0 and 1, where a higher value indicates better performance in ranking predictions.
- Hit@$N$: Hit@$N$ measures the proportion of correct predictions within the top $N$ ranked results. For example, when $N = 1$, we are interested in whether the correct answer is predicted within the top-ranked result. When $N = 3$, we evaluate whether the correct answer is within the top three predictions. The value of $N$ can be chosen based on the specific task and dataset characteristics, commonly set to 1, 3, or 10. Hit@$N$ values also range between 0 and 1, representing the proportion of correct predictions.

## C.4 Link Predication on ICEWS18

The link prediction experimental results of ICEWS18 are displayed in Table 4. The proposed LLM-DA outperforms existing state-of-the-art benchmarks on ICEWS18 across all metrics. Moreover, GPT-NeoX [48] is an important baseline because it also introduces GPT as LLMs. However, our proposed method still improves all metrics on ICEWS18. For example, the proposed method obtains 15.6% improvement under Hit@10 on ICEWS18. These phenomena demonstrate that the dynamic adaptation strategy can effectively update LLM-generated rules with current data, ensuring that the rules consistently incorporate the most recent knowledge and better generalize predictions for future events.

Table 4: Link prediction results on ICEWS18 dataset. The best results are in bold and - means the result is unavailable. ♠ denotes the TKGR methods, ♣ represents the LLMs-based TKGR. The results of the models with † are derived from [22], and ‡ is from [23].

| Type | Models | Train | ICEWS18 | | | |
|---|---|---|---|---|---|---|
| | | | MRR | Hit@1 | Hit@3 | Hit@10 |
| ♠ | RE-NET† [43] | ✓ | 0.287 | 0.188 | 0.327 | 0.482 |
| | RE-GCN [44] | ✓ | 0.326 | 0.223 | 0.367 | 0.526 |
| | TLogic‡ [5] | ✓ | – | 0.205 | 0.340 | 0.485 |
| | TiRGN [46] | ✓ | 0.336 | 0.232 | 0.379 | 0.542 |
| ♣ | PPT [21] | ✓ | 0.266 | 0.169 | 0.306 | 0.454 |
| | Llama-2-7b-CoH‡ [23] | ✓ | – | 0.223 | 0.363 | 0.522 |
| | Vicuna-7b-CoH‡ [23] | ✓ | – | 0.209 | 0.347 | 0.536 |
| | GPT-NeoX [48] | ✗ | – | 0.192 | 0.313 | 0.414 |
| | Mixtral-8x7B-CoH† [22] | ✗ | 0.330 | 0.218 | 0.378 | 0.549 |
| | LLM-DA (TiRGN) | ✗ | **0.357** | **0.255** | **0.403** | **0.570** |

## C.5 Long Horizontal Link Prediction

Long horizontal link prediction generally predict the events happening at time point $t + k * \triangle T$, where $\triangle T$ denotes the interval between time points and $k$ controls how far we will predict. To perform the long horizontal link prediction, we adjust the integral length $\triangle T$ on ICEWS14 and ICEWS05-15 datasets. As described in Figure 6 and Figure 7, we report the results corresponding to different $\triangle T$ on ICEWS14 and ICEWS05-15 and compare the proposed LLM-DA with the strongest baselines RE-GCN [44] and TiRGN [46]. Experimental results shows that LLM-DA can achieve the best performance on different $\triangle T$ both datasets. This phenomenon demonstrates that LLMs can effectively learn meaningful rules, and the dynamic adaptation strategy can fully update the LLM-generated rules, allowing for better adaptation to the constantly changing dynamics within TKGs.

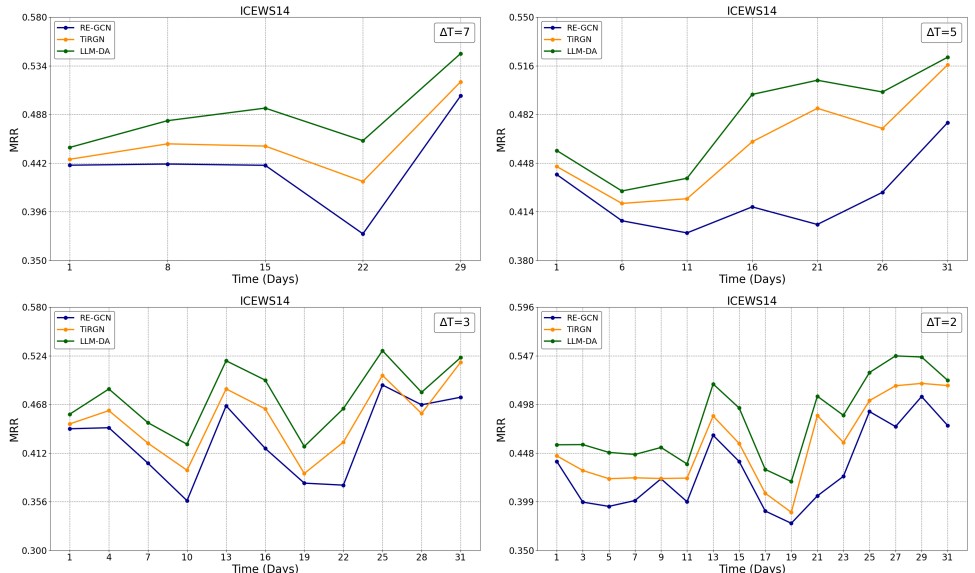

Figure 6: Long horizontal link forecasting: time-aware filtered MRR on ICEWS14 with respect to different time points.

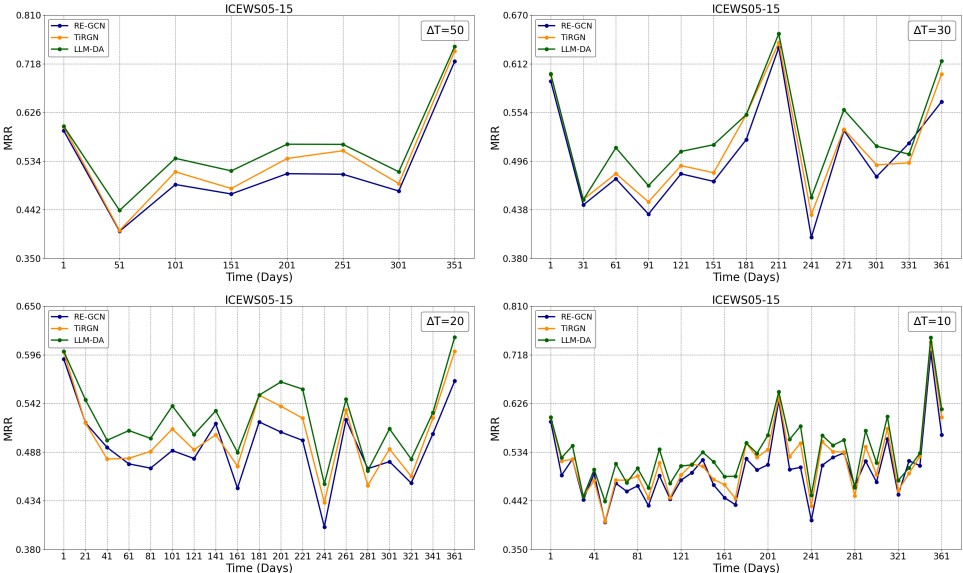

Figure 7: Long horizontal link forecasting: time-aware MRR performance on ICEWS05-15 with respect to different time points.

## C.6 Visualization Experiment

To visually present the changes of rule and candidate rankings during the *Rule Generation* and *Dynamic Adaptation*, we conducted a visualization experiment in Table 5. For the query *(China, Consult, ?, 2014-12-05)*, the actual answer **Japan** ranked fifth during the *Rule Generation* stage, but after the *Dynamic Adaptation*, it advanced to the top position. Meanwhile, the sequencing of rules in two phases also exhibits significant changes. For instance, the ranking of the rule "*Consult ← inv_Make_an_appeal_or_request & Express_···_meet_or_negotiate*" changes from fifth to first position. This highlights the efficacy of the dynamic adaptation strategy, which can update the LLM-generated rules with latest events to capture the evolving nature of TKGs.

Table 5: Visualization experiment for query *(China, Consult, ?, 2014-12-05)*. *Rules for Rule Generation* and *Rules for Dynamic Adaptation* represent the dynamic changes of the rules during the *Rule Generation* and *Dynamic Adaptation*. *Candidates for Rule Generation* and *Candidates for Dynamic Adaptation* indicate the changes in the corresponding candidate rankings. "**Japan**"is the actual answer.

| Query | (China, Consult, ?, 2014-12-05) |
|---|---|
| Rules for Rule Generation | ① inv_Host_a_visit & Host_a_visit & inv_Discuss_by_telephone
② Meet_at_a_'third'_location
③ Express_intent_to_engage_in_diplomatic_cooperation & inv_Host_a_visit & inv_Consult
④ Consult & Express_intent_to_cooperate & Consult
⑤ inv_Make_an_appeal_or_request & Express_intent_to_meet_or_negotiate |
| Candidates for Rule Generation | ① Malaysia
② Cambodia
③ China
④ South_Korea
⑤ **Japan** |
| Query | (China, Consult, ?, 2014-12-05) |
| Rules for Dynamic Adaptation | ① inv_Make_an_appeal_or_request & Express_intent_to_meet_or_negotiate
② Consult & Express_intent_to_cooperate & Consult
③ Meet_at_a_'third'_location
④ Express_intent_to_engage_in_diplomatic_cooperation & inv_Host_a_visit & inv_Consult
⑤ inv_Host_a_visit & Host_a_visit & inv_Discuss_by_telephone |
| Candidates for Dynamic Adaptation | ① **Japan**
② France
③ South_Korea
④ South_Africa
⑤ Malaysia |

[1] **Express_intent_to_engage_in_diplomatic_cooperation** represents *Express_intent_to_engage_in_diplomatic_cooperation_(such_as_policy_support).*

## C.7 Parameter Analysis

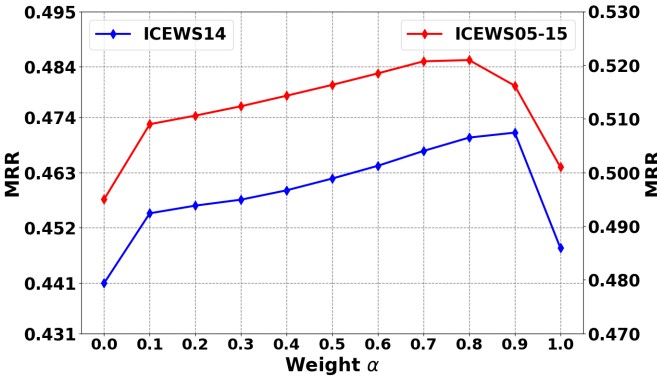

Figure 8: Comparison with different weight $\alpha$ on both datasets.

To experimentally study the effect of the weight $\alpha$ on ICEWS14 and ICEWS05-15, we tune the weight in a range $\{0, 0.1, \cdots, 1\}$ and observe the link prediction experimental results. As shown in Figure 8, the MRR performance exhibits a trend of initially increasing and then decreasing, reaching the peak at $\alpha = 0.9$ for ICEWS14, and $\alpha = 0.8$ for ICEWS05-15. These observations underscore that the weight $\alpha$ plays a crucial role in influencing the performance of LLM-DA. Specifically, the weight

$\alpha$ shows better performance at higher values, highlighting the importance of rule-based reasoning module.

Moreover, when $\alpha = 1$, indicating that only rules are used to generate candidates, the MRR performance decreases. This demonstrates that although LLMs can learn meaningful rules, the rules generated based on historical data cannot fully fit the future data due to the difference in distribution. This results in a few queries failing to find matching candidates when relying solely on rules for candidate generation.

## D   Limitations

Our limitations can be summarized as follows: (1) LLM-DA does not consider the semantics of nodes, which may reduce the quality of the sampled rules. (2) LLM-DA does not generate rules based on queries, which may result in the rules generally lacking specificity. (3) LLM-DA requires manually constructed prompts, which means that prompts need to be redesigned for different datasets. Moreover, manual construction of prompts also incurs significant costs. In the future, we will employ automated prompt learning to generate query-dependent rules through LLMs. Furthermore, we integrate the semantics of nodes during the temporal logical rules sampling stage.

