# OpenReview forum: "Large Language Models-guided Dynamic Adaptation for Temporal Knowledge Graph Reasoning"
_NeurIPS.cc/2024/Conference — NeurIPS 2024 poster_

### Official Review · Reviewer_CXWD · 2024-06-17

**Soundness:** 2
**Presentation:** 2
**Contribution:** 3
**Rating:** 5
**Confidence:** 4

**Summary:**

This paper proposes Large Language Models-guided Dynamic Adaptation (LLM-DA) to leverage LLMs to extract temporal logical rules for TKGR. Experimental results demonstrate LLM-DA significantly improves reasoning accuracy without the need for fine-tuning the LLM.

**Strengths:**

The paper is well-structured and easy to follow, the idea of extract and ground logic rules are clear motivated and illustrated.

Levearging LLM to exploit rules rather than predict the answers is interesting.

LLM-DA significantly outperforms the LLM-based methods on several datasets.

**Weaknesses:**

The writing could be improved. Some figures should be polished.

LLM-DA leverages GPT 3.5 as the LLM model, while the other baselines use only 7B LLM models. The authors should also conduct experiments to study the impact of different LLM models.

**Questions:**

Please see Weaknesses.

---

> ### Author Rebuttal · Authors · 2024-08-06
>
> # Comment for Reviewer CXWD:
>
> Thank you very much for your professional review and valuable suggestions. We have carefully considered and responded to the questions you raised.
>
> $\color{blue}{W.1:}$ The writing could be improved. Some figures should be polished.
>
> $\color{blue}{Re:}$ We really appreciate your comments. We will thoroughly revise the manuscript to improve the overall writing quality. Additionally, we'll review and refine all figures to ensure clarity and visual appeal.
>
> $\color{blue}{W.2:}$ LLM-DA leverages GPT 3.5 as the LLM model, while the other baselines use only 7B LLM models. The authors should also conduct experiments to study the impact of different LLM models.
>
> $\color{blue}{Re:}$ Thank you very much for your professional comments. We first want to point out that there are some larger LLMs (e.g., GPT-NeoX: 20B, and Mixtral-8x7B: 56B) selected as baselines for fair comparison. To further demonstrate the flexibility of our method, we conduct experiments about using different LLMs (Qwen 1.5 chat (7B) and Qwen 1.5 chat (72B)) in our proposed method. The experimental results on both datasets are shown as follows:
>
> | ICEWS14 | MRR | Hit@1 | Hit@3 | Hit@10 |
> |-|-|-|-|-|
> | Qwen 1.5 chat (7B) (TiRGN) | 0.461 | 0.353 | 0.506 | 0.660 |
> | Qwen 1.5 chat (72B) (TiRGN) | 0.462 | 0.356 | 0.508 | 0.661 |
>
> | ICEWS05-15 | MRR | Hit@1 | Hit@3 | Hit@10 |
> |-|-|-|-|-|
> | Qwen 1.5 chat (7B) (TiRGN) | 0.510 | 0.402 | 0.569 | 0.716 |
> | Qwen 1.5 chat (72B) (TiRGN) | 0.509 | 0.403 | 0.571 | 0.716 |
>
> For your convenience, we list the experimental results of baselines and the original results (using ChatGPT 3.5 as LLM) as follows:
>
> | | | | ICEWS14 | | -| | | ICEWS05-15 | |
> |-|-|-|-|-|- |-|-|-|-|
> | Models | MRR | Hit@1 | Hit@3 | Hit@10 |- | MRR | Hit@1 | Hit@3 | Hit@10 |
> |Llama-2-7b-CoH (7B) |-| 0.349 | 0.470 | 0.591 | -|-| 0.386 | 0.541 | 0.699 |
> |Vicuna-7b-CoH (7B) |-| 0.328 | 0.457 | 0.656 |- |-| 0.392 | 0.546 | 0.699 |
> |GPT-NeoX  (20B)  | - |0.334 | 0.460 | 0.565 |- |-|-|-|-|
> |Mixtral-8x7B-CoH  (56B)  | 0.439 | 0.331 | 0.496 | 0.649 |- | 0.497 | 0.380 | 0.564 | 0.713 |
> |-|-|-|-| |-|-|-|-|-|
> | Ours (ChatGPT 3.5 (TiRGN)) | 0.471 | 0.369 | 0.526 | 0.671 | -| 0.521 | 0.416 | 0.586 | 0.728 |
>
> From the above tables, it is evident that both the new LLMs, Qwen 1.5 chat (7B) and Qwen 1.5 chat (72B), perform better than the baselines, and achieve comparable performance to our proposed method (ChatGPT-3.5). This demonstrates that the proposed dynamic adaptation mechanism effectively captures temporal evolution patterns, allowing the generated rules to generalize better to future data.
> This validation across multiple model scales and architectures underscores the robustness of our approach, confirming its capability to leverage different LLMs effectively.
>
> In the final version, we will include additional experiments and provide a more comprehensive analysis to examine the impact of different LLMs.

---

> > ### Author Response · Authors · 2024-08-13
> > **Response to Reviewer CXWD**
> >
> > We sincerely appreciate your professional suggestions and look forward to your feedback on our rebuttal. Thank you once again for your valuable comments.

---

### Official Review · Reviewer_thzY · 2024-07-10

**Soundness:** 4
**Presentation:** 4
**Contribution:** 3
**Rating:** 7
**Confidence:** 5

**Summary:**

This paper explores the use of Large Language Models (LLMs) for Temporal Knowledge Graph Reasoning (TKGR). Specifically, the paper leverages LLMs for rule-based TKGR to identify temporal patterns and enable interpretable reasoning. Additionally, it introduces a dynamic adaptation strategy that iteratively updates the LLM-generated rules with the latest events, enhancing the model's adaptation to the evolving dynamics of TKGs.

**Strengths:**

S1. Compared to other LLM-based TKGR approaches, this paper demonstrates a significant improvement without the need for fine-tuning LLMs.
S2. It introduces a novel dynamic adaptation strategy, guiding LLMs to capture temporal evolution patterns in TKGs by continuously updating the knowledge rather than the LLMs themselves.
S3. Intuitive figures and tables enhance the readability of the paper.

**Weaknesses:**

W1. There is limited analysis of the constrained Markovian random walks.
W2. While Figure 1 intuitively reflects the paper's motivation, it could be improved. Moreover, the dynamic adaptation strategy, which should be the core focus, occupies a relatively small proportion in Figure 2.
W3. Several minor grammatical and expression issues need attention.

**Questions:**

Q1. Do the other modules in the paper, such as the contextual relation selector and the graph-based reasoning function, also require pre-training in addition to the LLMs?
Q2. The current experimental analysis is too superficial. To better validate the proposed method, a more in-depth analysis of the experiments is needed.

**Limitations:**

The authors clearly point out that this work is not query-dependent, which can lead to a lack of specificity in the generated rules. This provides a solid direction for future work to enhance the specificity of generated rules. Additionally, it is recommended that the authors discuss in depth the problems posed by not considering node semantics. Failure to consider node semantics may result in the loss of important contextual information and affect the overall accuracy of the model.
A more detailed discussion of this limitation would greatly benefit the paper.

---

> ### Author Rebuttal · Authors · 2024-08-03
>
> # Comment for Reviewer thzY:
>
> # Weaknesses:
>
> $\color{blue}{W.1:}$ There is limited analysis of the constrained Markovian random walks.
>
> $\color{blue}{Re:}$ Thanks for your suggestions. In the paper, we have provided theoretical analysis of the constrained Markovian random walks in Appendix A from the aspects of "Impact of Time Weighting" and "Traversal Properties". Specically, the time weighting enhances the model's sensitivity to recent data, improving its ability to capture short-term changes, while traversal properties ensure that the exploration of paths respects temporal continuity, leading to more accurate insights.
>
> $\color{blue}{W.2:}$ While Figure 1 intuitively reflects the paper's motivation, it could be improved. Moreover, the dynamic adaptation strategy, which should be the core focus, occupies a relatively small proportion in Figure 2.
>
> $\color{blue}{Re:}$ We appreciate your kind suggestion. We will refine the Figure 1 and Figure 2 in the final manuscript.
>
> $\color{blue}{W.3:}$ Several minor grammatical and expression issues need attention.
>
> $\color{blue}{Re:}$ Thank you for pointing out the minor grammatical and expression issues in our manuscript. We will thoroughly review and correct these issues in the final manuscript to ensure clarity and readability.
>
> # Questions:
>
> $\color{blue}{Q.1:}$ Do the other modules in the paper, such as the contextual relation selector and the graph-based reasoning function, also require pre-training in addition to the LLMs?
>
> $\color{blue}{Re:}$ Thank you for your insightful question. Both the contextual relation selector and the graph-based reasoning function are freezed without any tuning in our framework.
>
> $\color{blue}{Q.2:}$ The current experimental analysis is too superficial. To better validate the proposed method, a more in-depth analysis of the experiments is needed.
>
> $\color{blue}{Re:}$ Thank you for your suggestions. To further evaluate the proposed method, we explore the generalizability of our framework with different LLMs. Specifically, we replace the closed-source ChatGPT with two open-source LLMs of different sizes. (e.g., Qwen-1.5-chat-7B and Qwen-1.5-chat-72B). The detailed results can be found in our response to *reviewer CXWD. W2*. For your convenience, we compile the results showing the impact of different LLMs on experimental performance as follows:
>
> | ICEWS14 | MRR | Hit@1 | Hit@3 | Hit@10 |
> |-|-|-|-|-|
> | Qwen 1.5 chat (7B) (TiRGN) | 0.461 | 0.353 | 0.506 | 0.660 |
> | Qwen 1.5 chat (72B) (TiRGN) | 0.462 | 0.356 | 0.508 | 0.661 |
>
> | ICEWS05-15 | MRR | Hit@1 | Hit@3 | Hit@10 |
> |-|-|-|-|-|
> | Qwen 1.5 chat (7B) (TiRGN) | 0.510 | 0.402 | 0.569 | 0.716 |
> | Qwen 1.5 chat (72B) (TiRGN) | 0.509 | 0.403 | 0.571 | 0.716 |
>
> Experiment results show that the proposed method is flexible with the used LLMs showing the generalizability of our framework. Meanwhile, this provides deeper insights into the selection of LLMs and further validate the significant advantages of our proposed method in handling temporal data.
>
> In the final version, we will add more detailed experimental analysis to thoroughly demonstrate the robustness and effectiveness of our proposed method.

---

> > ### Comment · Reviewer_thzY · 2024-08-11
> >
> > The authors have addressed well my concerns, and, considering the comments from other reviewers, I would like to improve my rating.

---

> > > ### Author Response · Authors · 2024-08-11
> > > **Response to Reviewer thzY**
> > >
> > > Thank you for your timely reply! We sincerely appreciate your support for our work.

---

### Official Review · Reviewer_9wEQ · 2024-07-17

**Soundness:** 3
**Presentation:** 3
**Contribution:** 3
**Rating:** 6
**Confidence:** 3

**Summary:**

This paper introduces Large Language Models-guided Dynamic Adaptation (LLM-DA), a novel approach for Temporal Knowledge Graph Reasoning (TKGR). LLM-DA leverages LLMs to extract temporal logical rules from historical data, providing interpretable reasoning. It also incorporates a dynamic adaptation strategy to update these rules with the latest events, ensuring the extracted rules reflect the most recent knowledge.  Experimental results over several common datasets show that , LLM-DA, without the need of fine-tuning, significantly outperforms graph-based TKG method and LLM-based method in accurate reasoning.

**Strengths:**

- Though recently many efforts have been put into applying LLMs in TKG, this paper poses a novel way of applying LLM in TKGR as a temporal logic rule extractor to dynamically extract and update meaningful temporal patterns and complex temporal dependencies from the evolving event data, which provides extra interpretability to the forecasting process compare to a simple in-context LLM forecasting.
- The experiments have shown that the performance of the proposed method has surpassed the traditional graph-based TKG method and the recent LLM-based TKG methods.
- The analysis on the dynamic adaptation is very interesting and is in-depth and solid, shows the effectiveness of the dynamic adaptation in extracting the evolving temporal patterns.

**Weaknesses:**

- Long-horizon forecasting concerns: Concerns will be raised about the dynamic adaptation in cases where no near historical data is available, i.e. in the long-horizon forecasting task where the future data patterns and rules may largely differ from the seen data, how generalizable and robust this proposed method will be?
- Limited evaluation of generated rules:  While the LLM mainly participates in the rule extraction part, the paper lacks a comprehensive quantitative and qualitative evaluation of the generated rules. For instance, there's no assessment of how well the LLM generates symbolically and logically correct rules.

**Questions:**

Please refer to the questions raised in Weakness section.

**Limitations:**

Yes, the author has discussed the limitations in Appendix D, and proposed reasonable solutions fas the future works.

---

> ### Author Rebuttal · Authors · 2024-08-06
>
> # Comment for Reviewer 9wEQ:
> Thank you very much for your professional review and valuable suggestions. We have carefully considered and responded to the questions you raised.
>
> $\color{blue}{W.1:}$ Long-horizon forecasting concerns.
>
> $\color{blue}{Re:}$ Thank you for your insightful comments regarding the long-horizon forecasting. We want to clarify that we have conducted the long-horizon forecasting experiment in the paper, which can be found in Appendix C.5 (Figures 6 and 7) and Sec 5.3 RQ2 (Figure 4).
>
> -**Long-horizon Forecasting (time point):** To verify that the proposed method can generalize to diverse further distributions, we conduct the long horizontal link prediction in Appendix C.5. In this experimental setup, the model is adapted using only data available up to time point $t$, and then employed for predictions at subsequent time points $t+k\triangle T$ without any further adaptation. Here, $k$ controls how far we predict and $\triangle T$ denotes the interval between time points. In experiments, we vary $k$ to evaluate the performance over different horizons. Results depicted in Figures 6 and 7 demonstrate that our proposed LLM-DA maintains strong performance across various time points, indicating its robustness in dynamically adapting to new data patterns over long horizons.
>
> -**Long-horizon Forecasting (time interval):** We further evaluate the long-horizon forecasting results from the perspective of time intervals. Specifically, we divided the timeline into several intervals in chronological order, and these findings are thoroughly detailed in Section 5.3, RQ2. As shown in Figure 4, our method consistently outperforms existing methods across different time intervals, which validates the effectiveness of using LLMs to capture the evolving patterns of temporal knowledge.
>
> In the final version, we would add more details about the experiment settings and clarify the confusion.
>
> $\color{blue}{W.2:}$ Limited evaluation of generated rules.
>
> $\color{blue}{Re:}$ We appreciate your point about evaluating the generated rules. Given the large number of generated rules, a comprehensive manual evaluation is impractical. Instead, we use the quantitative metric *Support* to evaluate the quality of rule $\rho$, which represents the number of facts in KGs that satisfy the rules. *Support* is important for assessing rules, as it shows how often the facts can be inferred by the rules within KGs.
>
> Specifically, the rule $(\rho)$ defines the relation between two entities $e_s$ and $e_o$ at timestamp $t_l$,
>
> $\rho: = r(e_s, e_o, t_l) \leftarrow \wedge_{i=1}^{l-1}r^*(e_s,\ e_o,\ t_i),$
>
> where the left-hand side denotes the rule head with relation $r$ that can be induced by ($\leftarrow$) the right-hand rule body. The rule body is represented by the conjunction ($\wedge$) of a series of body relations $(r^* \in$ {$r_1,...,r_{l-1}$}).
>
> *Support* $s_\rho$ denotes the number of facts in Temporal Knowledge Graphs (TKGs) that satisfy the rule $\( \rho \)$, which is formally defined as:
>
> $s_\rho := (e_s, e_o, t_l) : \exists\wedge_{i=1}^{l-1}r^*(e_s,\ e_o,\ t_i) : \text{rule}$_$\text{body}(\rho) \wedge (e_s, r, e_o, t_l) \in \mathcal{G},$
>
> where $\wedge_{i=1}^{l-1}r^*(e_s,\ e_o,\ t_i)$ denotes a series of facts in KGs that satisfy the rule body $\(\text{rule}$_$\text{body}(\rho)\)$ and $\(e_s, r, e_o, t_l\)$ denotes the fact satisfying the rule head $\(r\)$.
>
> Following the *Support* metric, we can evaluate the quality of the rule $\rho$, and the results are shown as follows:
>
> | Datasets | Total Rules | Avg. Rules/$r$ | $s_\rho(Train)$ | $s_\rho(Test)$ | $s_\rho(ALL)$ | $s_\rho(Train)$ (filtered) | $s_\rho(Test)$ (filtered) |$s_\rho(ALL)$ (filtered) |
> |-----------------|-----------------|-----------------|-----------------|-----------------|-----------------|-----------------|-----------------|-----------------|
> | ICEWS14 | 31835 | 69.21 | 23.52 | 10.63 | 24.76 | 32.81 | 13.79 | 34.93 |
> | ICEWS05-15 | 41692 | 83.05 | 58.66 | 29.72 | 61.39 | 81.11 | 42.73 | 83.68 |
>
> In this Table, $s_\rho$ is the generated rules of LLMs, $s_\rho$ (filtered) denotes the filtered rules in Section 4.4, Eq. (8), which is the final rules used for rule application part; and $s_\rho(ALL)$ indicates the generated rules of LLMs are applied in the whole dataset.
>
> From the results, we can see that rules generated by LLM achieve a high support, indicating the good quality of them. In the final version, we will add experiments for rule evaluation in the appendix and conduct a comprehensive analysis.

---

> > ### Comment · Reviewer_9wEQ · 2024-08-11
> >
> > Thanks the authors for the rebuttal, especially the evaluation of the generated rules addresses my original concern, please consider adding this to future editions of the paper.

---

> > > ### Author Response · Authors · 2024-08-11
> > > **Response to Reviewer 9wEQ**
> > >
> > > We sincerely appreciate your timely reply and support for our work. We will add the experiments evaluating the generated rules in future editions of the paper. Your suggestions have been invaluable in enhancing the quality of our research, and we are eager to incorporate these improvements. Thank you once again for your constructive comment and encouragement.

---

### Decision · Program_Chairs · 2024-09-25

**Decision:**

Accept (poster)

**Comment:**

This paper leverages LLMs to infer rules for Temporal Knowledge Graph Reasoning (TKGR) -- link prediction with TKGs.  More specifically;
1) it first uses constrained random works to discover initial rules;
2) then the rules and related relations with scores (accuracy) are fed to an LLM to generate rules with better quality/coverage;
3) furthermore a dynamic adaptation strategy -- repeat the rule discovery (random walk), scoring (accuracy) and rewrite (LLM) steps for new data

Experimental results over a few TKG datasets show that , LLM-DA, outperforms graph-based TKG method and LLM-based method in accurate reasoning.

This work demonstrates the benefit of generating explicit reasoning rules from LLMs compared to pure graphic based approach and pure LLM based approach.